# Cloth-Splatting: 3D Cloth State Estimation from RGB Supervision

**Alberta Longhini*[1], Marcel Büsching*[1], Bardienus P. Duisterhof[2],**
**Jens Lundell[1], Jeffrey Ichnowski[2], Mårten Björkman[1], Danica Kragic[1]**
[1]KTH Royal Institute of Technology [2]Carnegie Mellon University

**Abstract:** We introduce Cloth-Splatting, a method for estimating 3D states of cloth from RGB images through a prediction-update framework. Cloth-Splatting leverages an action-conditioned dynamics model for predicting future states and uses 3D Gaussian Splatting to update the predicted states. Our key insight is that coupling a 3D mesh-based representation with Gaussian Splatting allows us to define a differentiable map between the cloth's state space and the image space. This enables the use of gradient-based optimization techniques to refine inaccurate state estimates using only RGB supervision. Our experiments demonstrate that Cloth-Splatting not only improves state estimation accuracy over current baselines but also reduces convergence time by $\sim 85\%$. Code and videos available at: kth-rpl.github.io/cloth-splatting.

**Keywords:** 3D State Estimation, Gaussian Splatting, Vision-based Tracking, Deformable Objects

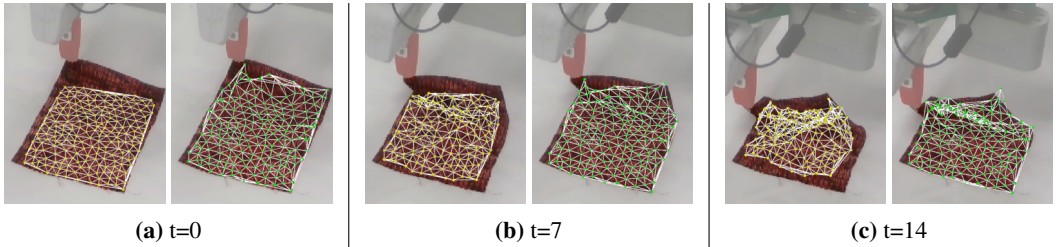

**(a)** t=0      **(b)** t=7      **(c)** t=14

**Figure 1: Cloth-Splatting state estimation of a real-world cloth.** We introduce Cloth-Splatting, a 3D state estimation method for cloth from sparse RGB observations that combines Graph Neural Network (GNN) with Gaussian Splatting (GS). Given an initial sequence of 3D mesh predictions from the GNN (left), Cloth-Splatting (right) updates it using RGB observations, achieving state-of-the-art 3D tracking performance for deformables.

## 1 Introduction

Teaching robots to fold, drape, or manipulate deformable objects such as cloths is fundamental to unlock a variety of applications ranging from healthcare to domestic and industrial environments [1]. While considerable progress has been made in rigid-object manipulation, manipulating deformables poses unique challenges, including infinite-dimensional state spaces, complex physical dynamics, and state estimation of self-occluded configurations [2]. Specifically, the problem of state estimation has led existing works on visual manipulation to either rely exclusively on 2D images, overlooking the cloth's 3D structure [3, 4, 5], or to use 3D representations that neglect valuable information in RGB observations [6, 7, 8].

Prior work on cloth state estimation often relies on 3D particle-based representations derived from depth sensors, including graphs [9, 10] and point clouds [11]. While point clouds effectively capture the object's observable state, they lack comprehensive structural information [6]. Alternative

---

\* Equal contribution. Correspondence to {albertal, busching}@kth.se

8th Conference on Robot Learning (CoRL 2024), Munich, Germany.

approaches rely on Graph Neural Networks (GNNs) trained on simulated data to predict graph representations using action-conditioned cloth dynamics [7, 8, 12]. However, these representations require online fine-tuning to bridge the sim-to-real gap. Using depth information to supervise the refinement of the cloth's 3D state estimates involves time-consuming simulation-based optimization techniques, as it lacks visual clues such as texture [13]. How to leverage RGB observations, which contain such information, remains an open problem.

We propose *Cloth-Splatting*, a method to estimate 3D states of cloth from RGB supervision by combining GS [14] with an action-conditioned dynamics model. The key idea of our method is to represent the 3D state of the cloth as a mesh and create a differentiable mapping between the cloth state space and the observation space using GS. This is achieved by populating the mesh faces with 3D Gaussians and expressing their positions relative to the mesh vertices. Given this, we can address the problem of estimating the 3D state of the cloth using a prediction-update framework akin to Bayesian filtering. Starting with a previous state estimate and a known robotic action, Cloth-Splatting predicts the next state using a learned dynamics model of the cloth. This prediction is then updated using RGB observations, leveraging the rendering loss provided by GS, allowing the refinement of the state estimate using visual clues such as texture and geometry.

We experimentally evaluate the 3D state estimation of cloth in simulated and real-world environments. Results show that Cloth-Splatting outperforms all 2D and 3D baseline tracking techniques, being 57 % more accurate and $\sim 85$ % faster than the best-performing baseline. We further showcase how refining mesh-based representations enables closed-loop manipulation of cloths. Together, the results suggest that tracking deformable objects using only RGB observations is feasible, efficient, and accurate. The contributions of this work are 1) a mesh-constrained object-centric extension of GS; 2) a vision-supervised 3D state estimation for cloth with action-conditioned dynamics priors, and 3) experiments that suggest successful sim-to-real transfer.

## 2 Related Work

**Cloth Manipulation.** Manipulating cloth in robotics involves tasks like folding, smoothing, lifting, and inserting objects into deformable bags [15, 3, 4, 5]. Manipulation approaches can be categorized into model-free methods, mapping cloth states or sensory data to actions [16, 17, 18], and model-based methods, utilizing cloth models to devise manipulation strategies [19, 13, 20]. Model-based approaches often rely on 3D representations focusing on optimizing pick-and-place actions. However, optimizing manipulation between pick and place in a closed-loop manner is crucial for handling planning errors and object variability [21, 22, 23, 24, 6], but remains underexplored, largely due to the challenges of real-world cloth state estimation [13]. Our work addresses this gap, enabling feedback-loop manipulations using graph representations.

**Cloth State Estimation.** Cloth state estimation is a well-studied area in robotics, computer vision, and computer graphics [25, 26]. While vision-based methods for 2D and 3D cloth state estimation have been developed for static observations of cloths lying flat on a surface [27, 28, 29, 30], tracking these estimates over time is challenging due to the complex non-linear dynamics of deformables. A first solution to this problem proposes a self-supervised method that leverages an action-conditioned dynamics model of the cloth and test-time optimization to refine 3D state representations from point cloud observation [13]. However, this method requires aligning real-world states with simulated ones and disregards informative feedback provided by RGB observations. Unlike previous methods, our work refines the 3D cloth states using RGB feedback. This is made possible by a differentiable GS map linking 3D states to image observations.

**Vision-based tracking.** Tracking points in 2D image space over time is a widely studied problem [31, 32, 33, 34, 35]. However, 3D visual trackers [36, 37] have gained popularity more recently, as they better address the occlusion challenges inherent in 2D tracking. Extensions of NeRF [38] to non-static scenes [39, 40, 41, 42], already allow to track the 3D motion of dynamic scene content, but often lack applicability to real-world scenarios [43]. 3D Gaussian Splatting [14] based methods, like Dynamic 3D Gaussians (DynaGS) [44], track by explicitly modeling the position and

covariance of each Gaussian over time. In contrast, 4DGS [45] learns a continuous deformation field to track Gaussian displacements. MD-Splatting [46] builds on 4DGS for cloth tracking using a shadow network and physics-inspired regularization. DeformGS [47] extends on MD-Splatting by learning object-centric masks, and simplifying the regularization terms. However, these methods require dense observations (at least 50 cameras for MD-Splatting and DeformGS) and involve computationally costly per-scene optimizations (see Fig. 4).

**Mesh-constrained GS.** Similar to Cloth-Splatting, several methods combine meshes and 3D Gaussians for deformable object representation [48, 49, 50]. However, like PhysGaussian [51], they rely on explicitly modeled dynamics and lack refinement with visual observations, limiting their tracking capabilities.

## 3 Problem Formulation

The problem addressed in this paper is estimating the 3D state of a cloth $\mathbf{M}_{t+1}$ at time $t+1$ given the observations $\mathbf{Y}_{1:t+1}$ and the agent's actions $\mathbf{a}_{1:t}$. The state of the cloth is represented as an augmented mesh $\mathbf{M}_t = (\mathbf{V}_t, \dot{\mathbf{V}}_t, \mathbf{E}_t)$, where $\mathbf{V}_t \in \mathbb{R}^{N \times 3}$ are the vertices positions, $\dot{\mathbf{V}}_t \in \mathbb{R}^{N \times 3}$ are the velocity of the vertices, and $\mathbf{E}_t \in \mathbb{Z}_+^{L \times 2}$ the edges. Only $\mathbf{V}_t$ and $\dot{\mathbf{V}}_t$ are estimated while the edges $\mathbf{E}_t$, that form a triangular mesh structure for connectivity, remain constant, i.e., $\mathbf{E}_0 = \mathbf{E}_1 = \cdots = \mathbf{E}_t$. The observations $\mathbf{Y}_{t+1} = \{\mathbf{I}_{t+1}^0, \ldots, \mathbf{I}_{t+1}^K\}$ are a set of K multi-view RGB images $\mathbf{I}_{t+1}^k \in \mathbb{R}^{w \times h \times 3}$ with unique camera matrices $\mathbf{P} = \{\mathbf{P}^0, \ldots, \mathbf{P}^K\}$ where $\mathbf{P}^k \in \mathbb{R}^{4 \times 4}$. The actions $\mathbf{a}_t \in \mathbb{R}^3$ are Cartesian end-effector velocities.

We reframe this estimation problem as a Bayesian filtering problem, where the goal is to infer the joint posterior distribution $p(\mathbf{M}_{t+1}|\mathbf{Y}_{1:t+1}, \mathbf{a}_{1:t})$ recursively over time with a general prediction-update framework. The prediction step is assumed to have access to a transition probability function $p(\mathbf{M}_{t+1}|\mathbf{M}_t, \mathbf{a}_t)$, which allows to compute a prior $p(\mathbf{M}_{t+1}|\mathbf{Y}_{1:t}, \mathbf{a}_{1:t})$ of the state of the system at time $t+1$:

$$p(\mathbf{M}_{t+1}|\mathbf{Y}_{1:t}, \mathbf{a}_{1:t}) = \int p(\mathbf{M}_{t+1}|\mathbf{M}_t, \mathbf{a}_t) p(\mathbf{M}_t|\mathbf{Y}_{1:t}, \mathbf{a}_{1:t-1}) d\mathbf{M}_t, \tag{1}$$

provided the history of observations $\mathbf{Y}_{1:t+1}$ and the agent's actions $\mathbf{a}_{1:t}$. Given a new observation $\mathbf{Y}_{t+1}$, the update of the prior estimate is proportional to the product of the measurement likelihood $p(\mathbf{Y}_{t+1}|\mathbf{M}_{t+1})$ and the predicted state. Thus, the joint posterior distribution is obtained as follows:

$$p(\mathbf{M}_{t+1}|\mathbf{Y}_{1:t+1}, \mathbf{a}_{1:t}) = \frac{1}{\eta} p(\mathbf{Y}_{t+1}|\mathbf{M}_{t+1}) p(\mathbf{M}_{t+1}|\mathbf{Y}_{1:t}, \mathbf{a}_{1:t}), \tag{2}$$

where $\eta$ is a normalization constant, ensuring that the posterior distribution integrates to 1. The state estimation problem then reduces to solving (2).

## 4 Method

Solving the problem of cloth 3D state estimation through a prediction-update framework requires modeling two key components: the transition probability function $p(\mathbf{M}_{t+1}|\mathbf{M}_t, \mathbf{a}_t)$ for the prediction step and the measurement likelihood $p(\mathbf{Y}_{t+1}|\mathbf{M}_{t+1})$ for the update step.

To handle the non-trivial deformable objects dynamics, we approximate the transition probability function with a deterministic GNN $f_{\boldsymbol{\theta}}$ parameterized by $\boldsymbol{\theta}$, which is described in more detail in Section 4.1. This GNN allows us to predict the state of the cloth at time $t+1$, denoted as $\hat{\mathbf{M}}_{t+1}$.

For the update step, we model the measurement likelihood $p(\mathbf{Y}_{t+1}|\mathbf{M}_{t+1})$ using a measurement model $h(\mathbf{Y}_{t+1}|\mathbf{M}_{t+1})$, which maps the predicted state $\hat{\mathbf{M}}_{t+1}$ to the predicted observation $\hat{\mathbf{Y}}_{t+1}$. The likelihood is then expressed as a function of the measurement error $||\mathbf{Y}_{t+1} - \hat{\mathbf{Y}}_{t+1}||_2^2$. Still, modeling the highly non-linear measurement model $h$ that maps from the state space of the cloth to the image space is challenging. The key insight of our work is to approximate the measurement model using GS, such that $\mathbf{Y}_{t+1} \approx h_{\text{GS}}(\mathbf{M}_{t+1}, \mathbf{P})$. Details on $h_{\text{GS}}$ are provided in Section 4.2.

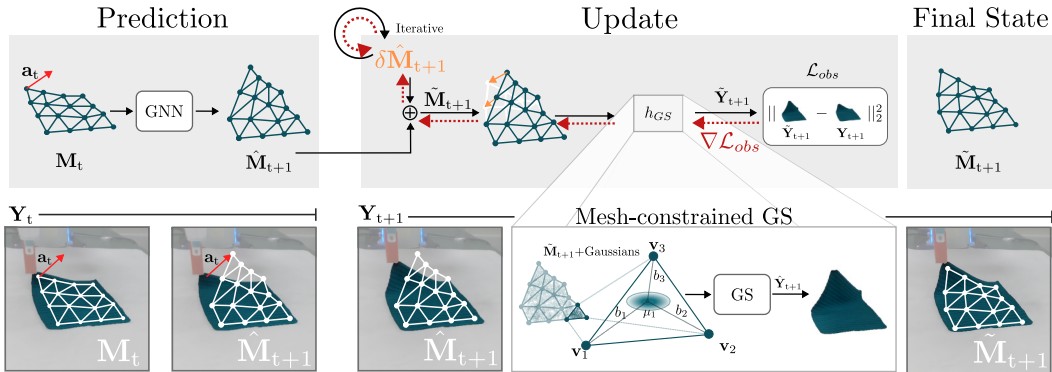

**Figure 2: Overview of Cloth-Splatting.** We estimate the 3D state of the cloth through a prediction-update framework. We use a GNN to approximate the transition function that estimates the next state $\hat{\mathbf{M}}_{t+1}$ given an action $\mathbf{a}_t$ and the state $\mathbf{M}_t$. We then employ our proposed mesh-constrained GS as a measurement model $h_{GS}$ to obtain an estimate of the observation $\tilde{\mathbf{Y}}_{t+1}$ given the current refined state $\tilde{\mathbf{M}}_{t+1} = \hat{\mathbf{M}}_{t+1} + \delta\hat{\mathbf{M}}_{t+1}$. The differentiability of GS allows us to iteratively optimize the update $\delta\hat{\mathbf{M}}_{t+1}$ of the state estimate via the photometric consistency loss between the ground-truth observation $\mathbf{Y}_{t+1}$ and the rendered state $\tilde{\mathbf{Y}}_{t+1}$.

By combining the transition model $f_{\boldsymbol{\theta}}$ and the measurement model $h_{\text{GS}}$, we create a differentiable mapping between the cloth's state space and the observation space. This allows us to obtain the updated state $\tilde{\mathbf{M}}_{t+1}$ via gradient-based optimization by minimizing the measurement error in the observation space:

$$\mathcal{L}_{obs} = ||\mathbf{Y}_{t+1} - h_{\text{GS}}(\tilde{\mathbf{M}}_{t+1}, \mathbf{P})||_2^2, \tag{3}$$

as described in Section 4.3.

An overview of the full method is visualized in Fig 2, showing the prediction-update framework proposed in this work.

## 4.1 State Prediction

We model the cloth's action-conditioned transition function $f_{\boldsymbol{\theta}}$ using a GNN parameterized by $\boldsymbol{\theta}$ that is designed to effectively handle the mesh structure and complex cloth dynamics [9]. The input to the GNN consists of the last $m$ states of the mesh $\mathbf{M}_{t-m:t}$ and the external robot action $\mathbf{a}_t$. To condition the GNN to the action, we assume the grasped particle is rigidly attached to the gripper, directly updating its position and velocity, similar to Wang et al. [52]. This constraint allows $f_{\boldsymbol{\theta}}$ to dynamically predict the next state of the cloth $\hat{\mathbf{M}}_{t+1} = f_{\boldsymbol{\theta}}(\mathbf{M}_{t-m:t}, \mathbf{a}_t)$ by propagating the effects of actions across the graph structure. Specifically, the network predicts the acceleration of each node, which we integrate using a forward-Euler integrator to compute the next velocity and position of each vertex. We train $f_{\boldsymbol{\theta}}$ on a one-step Mean Squared Error (MSE) loss between the predicted and ground-truth meshes.

## 4.2 Mesh-constrained GS

We utilize GS as the measurement model $\hat{\mathbf{Y}}_{t+1} = h_{\text{GS}}(\hat{\mathbf{M}}_{t+1}, \mathbf{P})$, where $h_{\text{GS}}$ is a differentiable map between the mesh representation and the image space.

**Background.** GS synthesises images by projecting the centers $\boldsymbol{\mu}$ and covariances $\boldsymbol{\Sigma}$ of 3D Gaussians to the image plane and then uses $\alpha$-blending to aggregate their colors $\mathbf{c}$ to pixel colors. To enforce the covariance matrix $\boldsymbol{\Sigma}$ to be positive and semi-definite, GS decompose it into a rotation $\mathbf{R}$ and scale $\mathbf{S}$ for each Gaussian:

$$\boldsymbol{\Sigma} = \mathbf{R}\mathbf{S}\mathbf{S}^{\text{T}}\mathbf{R}^{\text{T}}. \tag{4}$$

Given the camera matrix $\mathbf{P}$, the covariance matrix can be projected into image space as:

$$\boldsymbol{\Sigma}' = \mathbf{J}\mathbf{P}\boldsymbol{\Sigma}\mathbf{P}^{\text{T}}\mathbf{J}^{\text{T}}, \tag{5}$$

where $\mathbf{J}$ is the Jacobian of the affine approximation of the projective transformation. During rendering, the color $\mathbf{c}^p$ of a pixel is computed by blending $N$ ordered Gaussians overlapping the pixel:

$$\mathbf{c}^p = \sum_{i \in N} \mathbf{c}_i \alpha_i \prod_{j=1}^{i-1}(1 - \alpha_j), \tag{6}$$

where $\mathbf{c}_i$ is the color of each Gaussian and $\alpha_i$ is given by evaluating the projected 2D Gaussian with covariance multiplied with the learned opacity of each Gaussian.

**Mesh-constrained GS.** To connect GS to the mesh representation, we assign each Gaussian to a face of the initial mesh. The mean position $\boldsymbol{\mu}$ of each Gaussian is then defined in linear barycentric coordinates:

$$\mu = b_1\mathbf{v}^1 + b_2\mathbf{v}^2 + b_3\mathbf{v}^3, \tag{7}$$

where $\mathbf{v}^1, \mathbf{v}^2, \mathbf{v}^3 \in \mathbf{V}$ are the vertices of the assigned face and $b_1 + b_2 + b_3 = 1$. The rotation $\mathbf{R}$ of the Gaussian is then expressed relative to the assigned face. During a mesh deformation, the parameters of the Gaussians and the barycentric coordinates remain constant and represent the cloth's time-invariant appearance.

## 4.3 State Update

The goal of the state update is to improve the state prediction $\hat{\mathbf{M}}_{t+1}$ based on the observation $\mathbf{Y}_{t+1}$ such that the updated state $\tilde{\mathbf{M}}_{t+1}$ minimizes the measurement error (Eq. 3).

One option to refine the state would be to directly optimize the transition function $f_{\boldsymbol{\theta}}$ so it yields $\tilde{\mathbf{M}}_{t+1}$ as an improved state estimate. However, $f_{\boldsymbol{\theta}}$ requires recursive roll-out for estimating future states. Coupled with the internal propagations of the GNN, this can lead to vanishing gradients.

To circumvent the previous issues, we instead define the updated state as the sum:

$$\tilde{\mathbf{M}}_{t+1} = \hat{\mathbf{M}}_{t+1} + \delta\hat{\mathbf{M}}_{t+1}, \tag{8}$$

where $\delta\hat{\mathbf{M}}_{t+1} = u_{\boldsymbol{\psi}}(t + 1)$ is a learned residual state update parameterized by $\boldsymbol{\psi}$. The residual map $u_{\boldsymbol{\psi}}$ is modeled as a Multi-layer Perceptron (MLP), mapping the time to an offset from the state prediction, and is optimized using gradient descent on $\mathcal{L}_{obs}$. Intuitively, this approach keeps the prior estimate from the GNN fixed, refining the cloth state by minimizing the discrepancy between the observed image and the rendered cloth state. The residual map is optimized per scene with the initial output set close to zero. We also add the regularization loss $\mathcal{L}_{\text{reg}} = \mathcal{L}_{\text{SSIM}} + \mathcal{L}_{\text{iso}} + \mathcal{L}_{\text{magn}}$, where $\mathcal{L}_{\text{SSIM}}$ is the Structural Similarity Index Measure (SSIM) loss [53], $\mathcal{L}_{\text{iso}}$ ensures an As-Rigid-As-Possible (ARAP) behavior [54, 13] in the cloth, and $\mathcal{L}_{\text{magn}}$ minimizes overall motion for numerical stability. Details on the regularization and the mesh-constrained GS can be found in the Appendix.

## 5 Experimental Evaluation

The primary objective of the experiments is to evaluate how fast and accurate Cloth-Splatting can estimate 3D cloth states from RGB images compared to other baselines. We further ablate different components of our method, provide qualitative results, and demonstrate the feasibility of using Cloth-Splatting for robotic cloth manipulation in both simulated and real-world scenarios.

### 5.1 Experimental Set-up

**Dataset Generation.** Due to the lack of ground-truth states in real-world scenarios, we generated a synthetic dataset composed of 75 different scenes to conduct quantitative evaluations. We included three different object categories: TOWEL, SHORTS, and TSHIRT with five mesh variations of each object and five different trajectories per variation. The mesh variations were generated following Lips et al. [55]. Each trajectory consists of random interactions with the cloths, performed using the NVIDIA Flex simulator [56, 57]. We recorded RGB-D images from 4 different camera views for each scene and time step, where the depth is only used for the baseline methods. We used

Blender [58] to render photo-realistic scenes. We further record robot actions and ground-truth cloth states.

**Action-conditioned learned dynamics and mesh initialization.** We implemented the action-conditioned models using the Graph Network Simulator (GNS) architecture proposed in [9], which consists of a GNN with a node encoder, a processor, and a decoder. The input to the model is a mesh reconstructed from a 3D point cloud of the cloth at t = 0. We use Delaunay triangulation to reconstruct the mesh [59]. The point cloud can be obtained from either depth observations or multi-view stereo reconstructions [60]. We trained the model exclusively on the TOWEL object and split the dataset into a training, validation, and test set.

**Gaussian Splatting and Residual Model.** For initialization, two randomly parameterized Gaussians are positioned on each face of the mesh, by sampling their barycentric coordinates from the distribution $\mathcal{N}(1/3, 0.05)$ and normalizing them so that they add up to 1. To model the appearance of the cloth, we first optimize the Gaussians using observations from $t = 0$ and the measurement loss $\mathcal{L}_{\text{obs}}$ without regularization for the first 1.5k iterations. Subsequently, we jointly optimize the Gaussians and the residual network for an additional 5.5k – 6.5k iterations for learning the residual deformation and previously unseen parts of the cloth, such as a differently colored backside.

**Update.** We implement two different update procedures. The ITERATIVE update predicts one step ahead, refines the prediction with GS, and then uses the refined state as input to the GNN for the next prediction. In contrast, ROLLOUT predicts future states by unrolling the GNN over the entire trajectory and then refines all states simultaneously with GS, resulting in a faster overall runtime. We apply ROLLOUT in the tracking experiments and ITERATIVE in the manipulation experiments. A runtime and accuracy comparison between the approaches is provided in the appendix.

## 5.2 Quantitative Results

To quantify the tracking performance, we use: *median trajectory error* (MTE) [61], which measures the distance between the estimated cloth and ground-truth tracks; *position accuracy* ($\delta$) [33], which measures the percentage of tracks within the pre-defined distance thresholds 10, 20, 40, 80, and 160 mm to the ground truth; and the *survival rate*, which assesses the average number of frames until the tracking error exceeds a predefined threshold [61], which we set to 50 mm.

We compare Cloth-Splatting to the following baselines: Dynamic 3D Gaussians (DynaGS) [44], which separately models the positions and rotations of each Gaussian; and MD-Splatting [46], which extends GS by projecting non-metric Gaussians into a metric space to track deformable objects better. We also include the unrefined GNN predictions to establish a performance baseline, as well as a 2D tracking method obtained by evaluating RAFT [62] on all views and then reporting only the trajectories from the view with the lowest MTE. We denote this as RAFT-Oracle as it has access to the ground-truth cloth state to select the best-performing view.

### 5.2.1 Comparative Tracking Evaluation

We assess the tracking accuracy of Cloth-Splatting across the 75 synthetic scenes against the baselines. The results in Table 1 show Cloth-Splatting outperforms the next best baseline MD-Splatting by 9.66 % in MTE on average, while also having the best overall position accuracy $\delta_{\text{avg}}$ and survival rate. All methods show a varying performance between the different scenes, with all methods consistently being the worst on SHORTS which contain the largest, self-occluded deformation of the cloth with an example shown in Fig. 3. Analysing the unrefined GNN predictions shows a competitive performance, even on the shapes which are out-of-distribution from the training data, indicating that the GNN generalizes across different geometries.

### 5.2.2 Model Ablation

We report ablation studies on the mesh-constrained component of Cloth-Splatting in Table 2 on a SHORTS scene. In the following, we refer to each ablation by the corresponding identifier, eq. (A1).

**Table 1: Quantitative evaluation.** Comparison of Cloth-Splatting and the baselines in tracking quality. We report the mean and the standard deviation per metric ($\mu \pm \sigma$) and mark the best result **bold**.

| Metric | Method | SHORTS | TOWEL | TSHIRT | Mean |
|---|---|---|---|---|---|
| 3D MTE $\downarrow$ [mm] | RAFT-Oracle [35] | 26.511 $\pm$ 33.854 | 17.951 $\pm$ 12.523 | 8.062 $\pm$ 14.990 | 18.324 $\pm$ 23.821 |
| | DynaGS [44] | 15.987 $\pm$ 17.972 | 7.352 $\pm$ 3.265 | 9.574 $\pm$ 4.324 | 10.924 $\pm$ 11.246 |
| | MD-Splatting [46] | 7.043 $\pm$ 9.265 | 1.755 $\pm$ 2.711 | **2.109** $\pm$ 2.907 | 3.635 $\pm$ 6.235 |
| | GNN | 17.388 $\pm$ 15.824 | 9.157 $\pm$ 7.292 | 14.826 $\pm$ 12.295 | 13.853 $\pm$ 12.674 |
| | Cloth-Splatting | **4.928** $\pm$ 5.240 | **1.703** $\pm$ 1.213 | 3.159 $\pm$ 2.818 | **3.284** $\pm$ 3.722 |
| 3D $\delta_{\text{avg}}$ $\uparrow$ | RAFT-Oracle [35] | 0.660 $\pm$ 0.166 | 0.661 $\pm$ 0.114 | 0.744 $\pm$ 0.133 | 0.683 $\pm$ 0.143 |
| | DynaGS [44] | 0.773 $\pm$ 0.141 | 0.829 $\pm$ 0.076 | 0.808 $\pm$ 0.083 | 0.804 $\pm$ 0.105 |
| | MD-Splatting [46] | 0.816 $\pm$ 0.098 | 0.870 $\pm$ 0.059 | 0.855 $\pm$ 0.082 | 0.847 $\pm$ 0.083 |
| | GNN | 0.720 $\pm$ 0.128 | 0.791 $\pm$ 0.076 | 0.731 $\pm$ 0.141 | 0.747 $\pm$ 0.121 |
| | Cloth-Splatting | **0.851** $\pm$ 0.075 | **0.879** $\pm$ 0.057 | **0.858** $\pm$ 0.080 | **0.862** $\pm$ 0.072 |
| Survial rate $\uparrow$ | RAFT Oracle [35] | 0.666 $\pm$ 0.170 | 0.734 $\pm$ 0.119 | 0.752 $\pm$ 0.132 | 0.715 $\pm$ 0.145 |
| | DynaGS [44] | 0.795 $\pm$ 0.154 | 0.860 $\pm$ 0.072 | 0.850 $\pm$ 0.092 | 0.835 $\pm$ 0.113 |
| | MD-Splatting [46] | 0.869 $\pm$ 0.085 | 0.910 $\pm$ 0.059 | 0.881 $\pm$ 0.082 | 0.887 $\pm$ 0.077 |
| | GNN | 0.771 $\pm$ 0.138 | 0.860 $\pm$ 0.081 | 0.770 $\pm$ 0.138 | 0.800 $\pm$ 0.128 |
| | Cloth-Splatting | **0.917** $\pm$ 0.067 | **0.927** $\pm$ 0.059 | **0.888** $\pm$ 0.084 | **0.910** $\pm$ 0.072 |

Using the state predictions of the GNN without the GS update (A1) results in a high tracking error, likely due to the previously unobserved shape of the cloth. Similarly, using Cloth-Splatting without the GNN (A2) and relying only on the initial mesh state for initialization leads to poor tracking accuracy, as the model must learn the entire deformation just from visual observations. Additionally, removing the regularization terms (A3) significantly reduces accuracy as it fails to enforce cloth structural properties. Consequently, parts of the cloth stretch into unnatural states to satisfy the visual observations rather than maintaining the underlying shape. Interestingly, reducing the number of camera views (A6, A7) does not degrade the performance as much as other ablations, although we experience a larger drop in performance for a monocular setup (A5).

**Table 2: Model ablation.**

| Ablation | 3D MTE [mm] |
|---|---|
| (A1) Only GNN | 21.552 |
| (A2) No GNN | 16.135 |
| (A3) No $\mathcal{L}_{reg}$ | 15.772 |
| (A4) No $\mathcal{R}_t$ | 10.799 |
| (A5) 1 view | 16.525 |
| (A6) 2 views | 9.535 |
| (A7) 3 views | 9.004 |
| Full (4 views) | **8.923** |

#### 5.2.3 Time Ablation

To quantify how the cloth model prior affects convergence time, we compare the MTE of Cloth-Splatting to the best-performing baseline throughout their training durations. The experiments were performed on an NVIDIA 4090 GPU and an Intel i9-14900K processor. Fig. 4 shows that Cloth-Splatting achieves faster convergence times than the MD-Splatting and overall lower tracking error. The efficiency of Cloth-Splatting not only reduces computational costs but also enables faster adaptation to new scenarios, making it a more practical choice for real-world online applications.

### 5.3 Qualitative Results

We present simulated qualitative rendering and tracking results in Fig. 3 and compare these to MD-Splatting. Both methods capture the underlying dynamics of the scene. Nevertheless, the tracking results of MD-Splatting are negatively influenced by occasional tracking of visual artifacts. Although high-quality renderings are not the goal of Cloth-Splatting, the results show that the general visual appearance of the observed cloth is reproduced faithfully.

To showcase the transferability of Cloth-Splatting to the real world, we qualitatively evaluate the tracking of a real cloth folded by a 7-DOF Franka Emika Panda robot, observed by 3 extrinsically calibrated RealSense d435 from which we only collect RGB images. Fig. 1 shows qualitative real world results for Cloth-Splatting and the GNN predictions. While the GNN predictions partially predict the correct mesh structure, they suffer from compounding errors from the roll-out predictions. Subsequently, Cloth-Splatting successfully updates the mesh to better model the cloth shape.

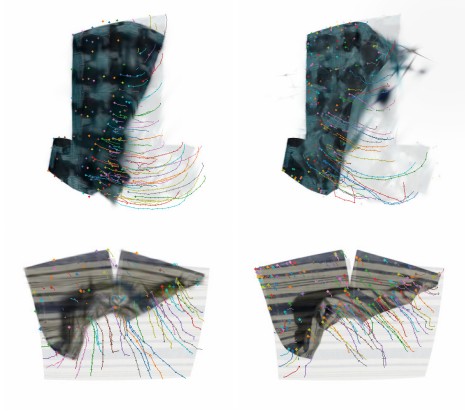

**(a)** Cloth-Splatting     **(b)** MD-Splatting

**Figure 3: Qualitative results (sim).** Tracking and rendering results on `SHIRT` (top) and `SHORTS` (bottom) scenes.

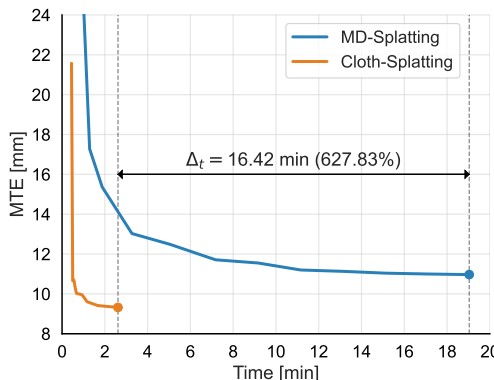

**Figure 4: Time ablation.** Convergence times of Cloth-Splatting and MD-Splatting on a `SHORTS` scene. The offset to the $y$-axis arises from the GNN predictions (1.36 s) and the 3D Gaussian initialization (26.55 s).

## 5.4 Robotic Manipulation Use Case

We further showcase that our state estimation process enables closed-loop optimization of folding trajectories using graph state representations. We focus on the half-folding task from Garcia-Camacho et al. [63], optimizing the trajectory between predefined pick-and-place positions. Our method refines the cloth's state estimate during each re-planning step, using model-predictive control (MPC) for planning and Cloth-Splatting for refinements (`MPC-CS`). We compare against a baseline using a predefined linear trajectory (`Fixed`), an open-loop (`MPC-OL`) baseline, and an oracle baseline (`OL-ORACLE`) with ground-truth state access. A detailed planning framework

**Table 3: Manipulation results** We report the mean and standard deviation of the MSE computed between the final cloth state and the goal state. Results are presented in units of $10^{-3}$.

| Method | TOWEL | TSHIRT |
|---|---|---|
| FIXED | $2.2 \pm 0.4$ | $2.4 \pm 0.4$ |
| MPC-OL | $1.8 \pm 2.1$ | $7.3 \pm 5.2$ |
| MPC-CS (us) | $0.6 \pm 0.6$ | $1.2 \pm 0.8$ |
| MPC-ORACLE | $0.4 \pm 0.2$ | $0.8 \pm 0.5$ |

is presented in Algorithm 2 in the Appendix. The results in Table 3 show that our method outperforms all baselines, achieving results comparable to those of the oracle, underscoring the effectiveness of our method in refining state estimates for model-based closed-loop manipulation. Qualitative results for simulation and real-world manipulations can be found in Appendix F.

## 6 Conclusions

We introduced Cloth-Splatting, an approach for 3D state estimation of cloths using RGB supervision. Cloth-Splatting integrates an action-conditioned cloth dynamics model with Gaussian Splatting to enhance the accuracy of 3D state representations based solely on RGB feedback using sparse (3-4) camera views. We experimentally validated the quality of these estimates and showcased the computational efficiency of our framework. These advancements position Cloth-Splatting as a more practical choice for real-world applications compared to existing baselines.

**Limitations.** While Cloth-Splatting shows significant improvements over the baselines, its speed is still insufficient for real-time applications. Additionally, the need for a calibrated multi-camera setup may present challenges in real-world scenarios, although scalable computer vision techniques for calibration, such as those in Dust3r [60], can help mitigate this issue. Since Cloth-Splatting assumes a static visual appearance, dynamic appearance changes, such as shadows or changing lighting conditions, can occasionally lead to the tracking of visual artifacts. Furthermore, the initial mesh initialization requires occlusion-free observations of the cloth. To address this limitation, recent advances in template-based reconstruction [64] of crumpled cloth present a promising direction for future research.

**Acknowledgments**

This work was supported by the Swedish Research Council; the Wallenberg Artificial Intelligence, Autonomous Systems and Software Program (WASP) funded by Knut and Alice Wallenberg Foundation; the European Research Council (ERC-884807); and the Center for Machine Learning and Health (CMLH). The computations were enabled by the Pittsburgh Supercomputing Center and by the Berzelius resource provided by the Knut and Alice Wallenberg Foundation at the Swedish National Supercomputer Centre.

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

# A  Cloth-Splatting Implementation

## A.1  Action-Conditioned Dynamics Architecture and Training

The action-conditioned dynamics model builds on the GNS architecture [9], which consists of three parts: encoder, processor, and decoder. The encoder consists of two MLPs, $\phi_p$ and $\phi_e$, which map vertices and edge features into latent embeddings $h_i$ and $g_{jk}$ respectively. The processor comprises $L = 15$ Graph Network (GN) blocks with residual connections that propagate the information throughout the mesh. Each GN block includes an edge update MLP, a vertice update MLP, and a global update MLP. The decoder is an MLP $\psi$ that outputs acceleration for each point: $\ddot{x}_i = \psi(h_i^L)$, which we use to update the position of each vertice of the cloth mesh via Euler integration.

The input vertice features consist of past $k = 3$ velocities and the vertice type. The vertice type is a binary flag used to distinguish grasped vertices from non-grasped vertices. The edge features include the distance vector $(v_j - v_k)$ and its norm $\|v_j - v_k\|$. To condition the model on the actions of the robot, we update the velocity of the pick point based on the robot's action before giving the state of the cloth in input to the network. This facilitates the propagation of the actions throughout the GNS to predict future states.

We train the action-conditioned dynamics on towel objects, using the mean-squared error between predicted and simulator-obtained accelerations for 200 epochs using Adam [65].

## A.2  Mesh-constrained Gaussian Splatting

**Orientation estimation**  The orientation of the Gaussian $\mathbf{R}$ depends on the orientation of the associated face $\mathbf{R}^\mathrm{F}$ and the static orientation of the Gaussian on the face $\mathbf{R}'$, resulting in $\mathbf{R} = \mathbf{R}^\mathrm{F}\mathbf{R}'$. We estimate the face orientation $\mathbf{R}^\mathrm{F}$ in a deformed mesh via a vector registration between the initial positions of the face vertices $\mathbf{v}_0^i$ and the positions in the deformed state $\mathbf{v}_t^i$. This can be formulated as an optimization problem

$$\min_{\mathbf{R}^\mathrm{F}} \sum_{i=\{1,2,3\}} ||\mathbf{R}^\mathrm{F}\mathbf{v}_0^i - \mathbf{v}_t^i||^2, \tag{9}$$

which we solve using the RoMa toolbox [66]. During the optimization of the barycentric coordinates, we permit them to take on negative values, which would locate the Gaussian outside of the assigned face. This approach enables us to detect, when a Gaussian should get assigned to a different face, with the barycentric coordinates then being relative to the vertices of that new face.

**Optimization**  For the mesh-constrained Gaussian Splatting, we build on the original Gaussian Splatting procedure, with the main modification that we constrain the Gaussian positions on the surface of a pre-defined mesh as described in the 4.2. Details of Gaussian Splatting, such as the pruning, densification, and regular resetting of opacities, remain unchanged. Nevertheless, in order to keep the number of 3D Gaussians low, we increase the required opacity for Gaussians to not be pruned, since we can assume that there are no transparent parts on the reconstructed cloth. Therefore, a normal reconstruction of the appearance of cloth only requires about 4k Gaussians.

We observe that when the Gaussians are optimized over the whole range of training, the visual appearance and the tracking degrades. For example, the Gaussian position on the mesh starts to fit the deformed appearance instead of the residual dynamics model learning the proper offset. Therefore, the learning rates of the Gaussians' attributes (color, position, scale, ...) are annealed over the first 6k iterations and afterward frozen so only the residual dynamics model is optimized.

## A.3  Residual dynamics model

We implement the residual dynamics model as a 3-layer ReLU MLP with a width of 256. The input to the MLP is a scalar value in the range $[0, 1]$, corresponding to the normalized time step, which is encoded with the sinusoidal frequency encoding also used in NeRF [38], using 6 frequencies. The output size is $3 \times N$, with $N$ being the number of vertices in the mesh.

We randomly initialize weights and biases of the output layer with a zero-centered normal distribution with a covariance of 0.0001, to start with a residual close to zero.

## A.4 Regularization

As discussed in Section 4.3, we learned the state updated by adding the following regularization losses: $\mathcal{L}_{\text{reg}} = \mathcal{L}_{\text{SSIM}} + \mathcal{L}_{\text{iso}} + \mathcal{L}_{\text{magn}}$, where $\mathcal{L}_{\text{SSIM}}$ is the SSIM loss [53], $\mathcal{L}_{\text{iso}}$ ensures neighboring vertices in the cloth maintain a constant distance, and $\mathcal{L}_{\text{magn}}$ minimizes overall motion.

The isometric loss:

$$\mathcal{L}_{\text{iso}} = \sum_{t=0}^{T-1} \sum_{i=0}^{N-1} \sum_{\mathcal{N}(v_{t,i})} |d(v_{t,i}, v_{t,j}) - d(v_{t+1,i}, v_{t+1,j})| \tag{10}$$

ensures that the neighbouring vertices $\mathcal{N}(v_{t,i})$ of $v_{t,i}$ maintain a constant distance from time $t$ to $t+1$.

The structural similarity index measure loss (SSIM) [67] is estimated for windows of the images and goes beyond the purely per-pixel color loss and also considers color gradients within the pixels local neighborhood. The loss between two windows $w$ an $v$ can be estimated with:

$$\mathcal{L}_{\text{SSIM}}(v, w) = \frac{(2\mu_v\mu_w + c_1)(2\sigma_{vw} + \mathbf{c}_2^p)}{(\mu_v^2 + \mu_w^2 + c_1)(\sigma_v^2 + \sigma_w^2 + c_2)}, \tag{11}$$

where $\mu$ is the mean color of each window, $\sigma^2$ the color (co-)variances, and $c_1$ and $c_2$ are constants to stabilize the loss.

The motion loss:

$$\mathcal{L}_{\text{magn}} = \sum_{t=0}^{T-1} \sum_{i=0}^{N-1} ||v_{t,i} - v_{t+1,i}||_2^2 \tag{12}$$

encourages to learn a solution with the smallest possible motion per vertice, which we found necessary to prevent instabilities during training.

# B Synthetic Data

The synthetic dataset consists of meshes representing three types of cloth objects: TSHIRT, SHORTS, and TOWEL. We procedurally generate meshes with random configurations, sizes, and overall shapes for each category based on the methods detailed in [55]. Post-generation, the meshes are deformed using NVIDIA Flex [56, 57] with random manipulation trajectories.

The manipulation trajectories are constructed using quadratic Bézier curves with three control points. Specifically, the pick and place locations represent the primary control points, which we randomly selected on the cloth particles. The third control point, positioned midway between the pick and place points, was set to a random height within the range $[0.05, 0.15]$ cm. Additionally, this control point was randomly tilted between $[-\pi/4, \pi/4]$ rad around the axis formed by the pick and place points to add variability in the manipulation trajectories. We finally discretized the manipulation trajectory into a series of small displacements depending on the gripper velocity, $\Delta x_1, \ldots, \Delta x_T$, ensuring:

$$x_{\text{pick}} + \sum_{i=1}^{T} \Delta x_i = x_{\text{place}},$$

randomly sampling the gripper velocity in the interval $[0.5, 2]$ cm/s.

To bridge the simulation-to-reality gap, we rendered the complete manipulation trajectory using Blender [58].

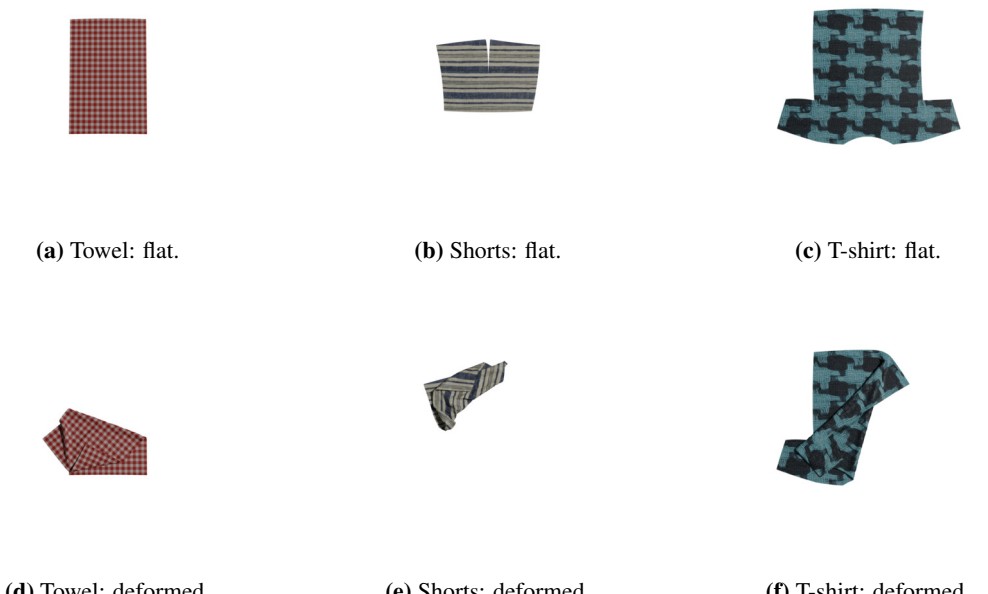

| **(a)** Towel: flat. | **(b)** Shorts: flat. | **(c)** T-shirt: flat. |

| **(d)** Towel: deformed. | **(e)** Shorts: deformed. | **(f)** T-shirt: deformed. |

**Figure 5:** Example of synthetic images generated for the objects considered in our experiments (towel, shorts, t-shirt). For each object, we show the flat (top row) and the deformed (bottom row) states, rendered with Blender.

## C   Real-world Set-up and Data Collection

The real-world set-up is shown in Fig. 6. We used 3 calibrated RealSense d435 cameras to collect RGB observations of the environment. We utilized one rectangular cloth for the experiments, also visualized in Fig. 6. The robot used for the experiments was a Franka-Emika Panda robot. We employed a Cartesian position controller to execute a folding trajectory, which was randomly generated using the same procedure as the simulated data. We assumed prior knowledge of the pick and place locations and that the cloth was already in a grasped configuration.

We recorded RGB observations from all three cameras throughout the manipulation process. Depth observations were additionally captured at $t = 0$ to initialize the cloth mesh for dynamics predictions. At each timestep, segmentation and video tracking modules pre-trained on Grounding-DINO [68] and Segment Anything (SAM) [69] were used to generate masks of the cloth and the gripper, respectively, using the prompts "cloth" and "robot gripper." These masks were subsequently tracked over time using the video tracker XMEM [70].

## D   Time-comparison between `ITERATIVE` and `ROLLOUT` updates

For the tracking part application of our method, we predict future states by unrolling the GNN for the full trajectory length and then refine all states via GS. In the following, we refer to this version of our method as `ROLLOUT`. A second option is to predict $H$-steps ahead, refine these steps with GS, and then use the latest refined states to update the input of the GNN for the next $H$ predictions. We will refer to this option as `ITERATIVE`. While `ROLLOUT` has the potential to be faster than `ITERATIVE`, it suffers from a larger error accumulation during the rollout of the GNN, which may negatively affect overall tracking performance. This experiment aims to evaluate the trade-off between iteratively updating the GNN input and the overall time of execution.

In Table 4, we present the tracking and execution time results over a single `TOWEL` scene. For the `ITERATIVE` method, we report the results for predicting 1, 2, 4, or 8 steps ahead before updating

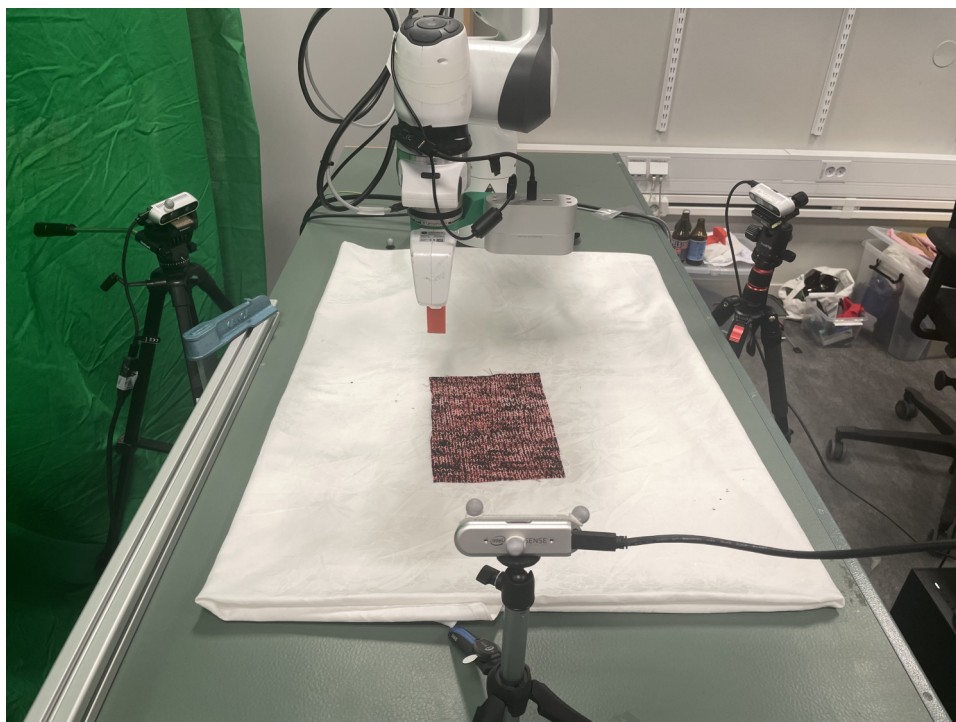

**Figure 6:** Overview of experimental set-up.

the GNN input with the refined states. `ROLLOUT` predicts and updates the full 16 timesteps of the scene jointly. Training parameters are the same for both methods, with the difference that we update the predictions `ITERATIVE` for 1000 iterations per prediction-update step, while we update the predictions from `ROLLOUT` directly 6000 iterations.

We can observe that doubling the number of prediction steps approximately halves the total time of `ITERATIVE`. The results clearly show the trade-off between tracking quality and time. Since one of our objectives was to improve the speed of computation, we chose to present in the paper the tracking results of the `ROLLOUT` method, as it offers a significant speed advantage while maintaining sufficient accuracy.

**Table 4:** Comparison of `ITERATIVE` and `ROLLOUT` versions of Cloth-Splatting.

|  |  | ITERATIVE |  |  |  | ROLLOUT |
| --- | --- | --- | --- | --- | --- | --- |
| prediction steps $H$ |  | 1 | 2 | 4 | 8 | 16 |
| 3D MTE [mm] | ↓ | 0.767 | 0.890 | 0.819 | 1.170 | 5.328 |
| 3D $\delta_{\text{avg}}$ | ↑ | 0.893 | 0.891 | 0.893 | 0.867 | 0.753 |
| Survival rate | ↑ | 0.937 | 0.950 | 0.947 | 0.907 | 0.776 |
| Time [min] | ↓ | 53:35 | 30:14 | 15:53 | 9:32 | 2:12 |

## E    Robustness Against Initialization Error

One of the assumptions of our method is to have access to an initial estimate of the mesh of the cloth at time $t = 0$. However, this estimate is prone to error due to sensor noise or approximation errors. To assess the robustness of our method to errors in mesh initialization, we selected a simulated towel and applied several augmentations to its initial mesh: `TRANS`, `ROT`, `SCALING`, `NOISE` and all combined `TRSN`. For each augmentation, we randomly sampled 10 different values. Given a cloth size of $0.2 \times 0.2 [\text{m}^2]$, we sampled $x$ and $y$ translations within $[-0.05, 0.05]$, $z$ translations

within $[-0.003, 0.003]$, yaw rotations within $[-30, +30]$ degrees, random scaling coefficients within $[0.8, 1.2]$, and additive random noise sampled from a multivariate Gaussian distribution with zero mean and 0.005 variance. These augmentations were applied to the initial mesh before being used to unroll the GNN. We then refined the predictions with our method and evaluated the tracking performance as in the previous section.

Building on this initial analysis, we expanded our evaluation by introducing a variation of the experiment that tests our method's ability to refine mesh initialization based on observations of the cloth's initial state. In this extended approach, we first refined the augmented mesh with our method using observations at time $t = 0$. Following this, we unrolled the GNN using the refined initial mesh. Lastly, we refined the predictions with our method and evaluated the tracking performance.

The results of these experiments are presented in Table 5. While the tracking error is the lowest on the error-free initialization, our method is still able to achieve comparable tracking accuracy despite the inaccurate mesh initialization. Refining the initial mesh state resulted in a modest improvement overall. However, it was crucial in preventing two initializations with ROT augmentations from producing tracking performance so poor that the evaluation script could not generate valid metrics (highlighted with $^\dagger$ in the Table. 5).

This brings us to the conclusion that, while an error on the initialization has a negative influence on the tracking quality, Cloth-Splatting remains capable of sufficiently tracking the cloth and is therefore not significantly limited by this factor.

**Table 5: Tracking with initialization error:** We report the tracking metrics as mean per augmentation type and include the error free initialization as **Ref**.

| Metric | Refined | Augmentation | | | | | | Ref |
|---|---|---|---|---|---|---|---|---|
| | | TRANS | ROT | SCALING | NOISE | TRSN | Mean | |
| 3D MTE | No | 3.346 | 2.598$^\dagger$ | 2.998 | 2.482 | 3.595 | 2.961 | 2.193 |
| $\downarrow$ [mm] | Yes | 3.123 | 2.887 | 2.763 | 2.859 | 3.394 | 2.953 | |
| 3D $\delta_{\mathrm{avg}}$ | No | 0.819 | 0.825$^\dagger$ | 0.823 | 0.826 | 0.815 | 0.822 | 0.835 |
| $\uparrow$ | Yes | 0.824 | 0.826 | 0.828 | 0.825 | 0.820 | 0.825 | |
| Survival | No | 0.865 | 0.871$^\dagger$ | 0.867 | 0.869 | 0.864 | 0.867 | 0.887 |
| $\uparrow$ | Yes | 0.871 | 0.872 | 0.875 | 0.870 | 0.866 | 0.872 | |

$^\dagger$: Two augmentations had to be excluded since their tracking performance was too insufficient to estimate the metrics.

## F  Manipulation Experiment Beyond State Estimation

In this experiment, we demonstrate both in simulation and in the real world that our state estimation process enables closed-loop optimization of folding trajectories using graph state representations. We focus on the benchmarking half-folding task introduced in [63], where the objective is to fold a cloth in half. We assume pick-and-place positions to be given a-priori, and we aim to optimize the folding trajectory between these two. This task is particularly challenging with only one gripper, as a predefined linear trajectory between the pick and place positions does not result in an accurate fold. We illustrate this in Figure 7 (right - Fixed), where we show that folding by executing a predefined fixed trajectory between the pick and the place locations results in a poor fold. We refer to this manipulation as Fixed baseline. This scenario, common in model-based cloth manipulation, underscores the importance of optimizing the folding trajectory beyond the pick-and-place locations to achieve the desired goal. To address this, we optimize the trajectory in a closed-loop manner, planning from a sequence of random actions using our prediction-update framework along with model-predictive control (MPC). During each re-planning step, we obtain the refined state estimate of the cloth using our method. The algorithm describing the state refinement is presented in Alg. 1, while an overview of the planning algorithm is presented in 2. We refer to this manipulation approach as (MPC-CS).

**Algorithm 1: Cloth-Splatting** - State refinement.

**Result:** Refined state $\tilde{\mathbf{M}}_{t+1}$.

**Input:** Estimated state: $\hat{\mathbf{M}}_{t+1}$, New observation: $\mathbf{Y}_{t+1}$, Measurement model: $h_{GS}$, Camera
matrices: $\mathbf{P}$, Epochs: $E$.

1  $\delta\hat{\mathbf{M}}_{t+1} \leftarrow 0$
2  $\tilde{\mathbf{M}}_{t+1} \leftarrow \hat{\mathbf{M}}_{t+1} + \delta\hat{\mathbf{M}}_{t+1}$
3  **for** $e \leftarrow 0$ **to** $E$ **do**
4      $\tilde{\mathbf{Y}}_{t+1} \leftarrow h_{GS}(\tilde{\mathbf{M}}_{t+1}, \mathbf{P})$
5      $\mathcal{L}_{obs} \leftarrow ||\mathbf{Y}_{t+1} - \tilde{\mathbf{Y}}_{t+1}||_2^2$                                ▷ Eq. (3)
6      $\delta\hat{\mathbf{M}}_{t+1} \propto \nabla\mathcal{L}_{obs}$
7      $\tilde{\mathbf{M}}_{t+1} \leftarrow \hat{\mathbf{M}}_{t+1} + \delta\hat{\mathbf{M}}_{t+1}$
8  **end**

**Simulation:** We consider one TOWEL and one TSHIRT to test our method on variations of cloth types. The success is evaluated by calculating the mean-squared error (MSE) between the desired goal state and the final state of the cloth. We compare MPC-CS against the following baselines: the FIXED baseline previously introduced, an open-loop (MPC-OL) baseline that plans the best actions in an open-loop fashion, and an oracle baseline that has access to the ground truth state of the cloth at each time step (OL-ORACLE). We repeat the folding task 10 times per method and report the results in Table 6. Our method outperforms all baselines, achieving results comparable to those of the oracle. Additionally, we present qualitative results of the final fold achieved by each method in Fig. 7 and Fig. 8 for the towel and the t-shirt respectively. These results underscore the effectiveness of our method in refining state estimates for model-based closed-loop manipulation. Although the planning is not fast enough to be executed in real-time, this is the first approach to showcase closed-loop manipulation with graph representations, underscoring the relevance of our proposed state estimation method.

**Real world:** We evaluate our method, MPC-CS, on real-world half-folding, comparing it against two baselines: FIXED and the open-loop MPC-OL. Unlike the simulation experiments, we lack ground-truth meshes for this evaluation. Figure 9 presents qualitative results for each method, with the final outcome depicted at Time 4. It is evident that the FIXED baseline, which does not optimize the folding trajectory, and the MPC-OL baseline, which operates in an open-loop manner, both fail to produce satisfactory folds. In contrast, our method successfully refines the mesh estimate and re-plans the fold, resulting in a superior final outcome. This experiment demonstrates the applicability of our state estimation method in manipulation tasks.

**Algorithm 2:** Closed-loop manipulation with Cloth-Splatting iterative update.

---

**Result:** Optimized folding actions $a^*_{0:T}$.
**Input:** Initial observation: $\mathbf{Y}_0$, Initial cloth point cloud: $\mathbf{PC}_0$, Camera matrices $\mathbf{P}$, Pick and place positions: $\{x_{\text{pick}}, x_{\text{place}}\}$ , Goal state: $\mathbf{M}_g$, Transition function: $f_\theta$, Measurement model: $h_{GS}$, Planning Horizon: $T$, Prediction Horizon: $H$, Number of action candidates: $N$, Initial control sequence $\mathbf{a}_{0:H}$, Control variance: $\Sigma$

1   $\mathbf{M}_0 \leftarrow$ Mesh Initialization($\{\mathbf{PC}_0\}$)
2   $\tilde{\mathbf{M}}_0 \leftarrow \mathbf{M}_0$
3   Initialize Gaussian baricenters $\leftarrow h_{GS}(\mathbf{M}_0, \mathbf{P})$
4   **for** $t \leftarrow 1$ **to** $T$ **do**
5      **for** $n \leftarrow 1$ **to** $N$ **do**
6         $\mathbf{a}^n_{t:t+H} \leftarrow \mathcal{N}(\mathbf{a}_{t:t+H}, \Sigma)$           ▷ Sample candidates
7         $\hat{\mathbf{M}}_t \leftarrow \tilde{\mathbf{M}}_t$           ▷ Refined state as input
8         **for** $h \leftarrow t$ **to** $t + H$ **do**
9             $\hat{\mathbf{M}}_{h+1} \leftarrow f_\theta(\hat{\mathbf{M}}_h, \mathbf{a}^n_h)$           ▷ Model rollout
10             $\mathcal{J}^h(\mathbf{a}^n_h) \leftarrow ||\hat{\mathbf{M}}_{h+1} - \mathbf{M}_g||^2_2$           ▷ Cost function
11         **end**
12         $\mathcal{J}^n(\mathbf{a}^n_{t:t+H}) \leftarrow \sum_{h=t}^{t+H} \mathcal{J}^h(\mathbf{a}^n_h)$
13      **end**
14      $a^*_{t:t+H} \leftarrow \underset{n\in\{1,...,N\}}{\arg\min} \; \mathcal{J}^n(\mathbf{a}^n_{t:t+H})$
15      Execute   $a^*_t$
16      $\mathbf{Y}_{t+1} \leftarrow$ New observation
17      $\tilde{\mathbf{M}}_{t+1} \leftarrow \text{CS}(\hat{\mathbf{M}}^{a^*}_{t+1}, \mathbf{Y}_{t+1}, h_{GS}, \mathbf{P})$           ▷ Cloth-Splatting - Alg.(1)
18 **end**

---

**Table 6:** Comparison of different manipulation strategies. Each method is tested 10 times, and we report the mean and standard deviation of the MSE computed between the final state and the goal state. Results are presented in units of $10^{-3}$.

| Metric | Object | FIXED | MPC-OL | MPC-CS (us) | MPC-ORACLE |
|--------|--------|-------|--------|-------------|------------|
| MSE | TOWEL | $2.2 \pm 0.4$ | $1.8 \pm 2.1$ | $0.6 \pm 0.6$ | $0.4 \pm 0.2$ |
| MSE | TSHIRT | $2.4 \pm 0.4$ | $7.3 \pm 5.2$ | $1.2 \pm 0.8$ | $0.8 \pm 0.5$ |

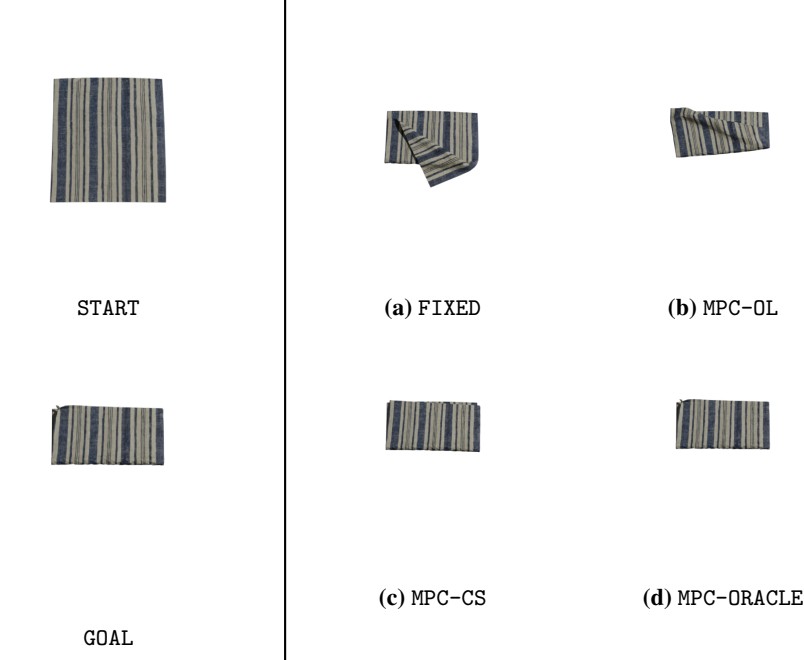

START     **(a)** `FIXED`     **(b)** `MPC-OL`

**(c)** `MPC-CS`     **(d)** `MPC-ORACLE`

GOAL

**Figure 7:** Qualitative results of the half folding for different manipulation strategies.

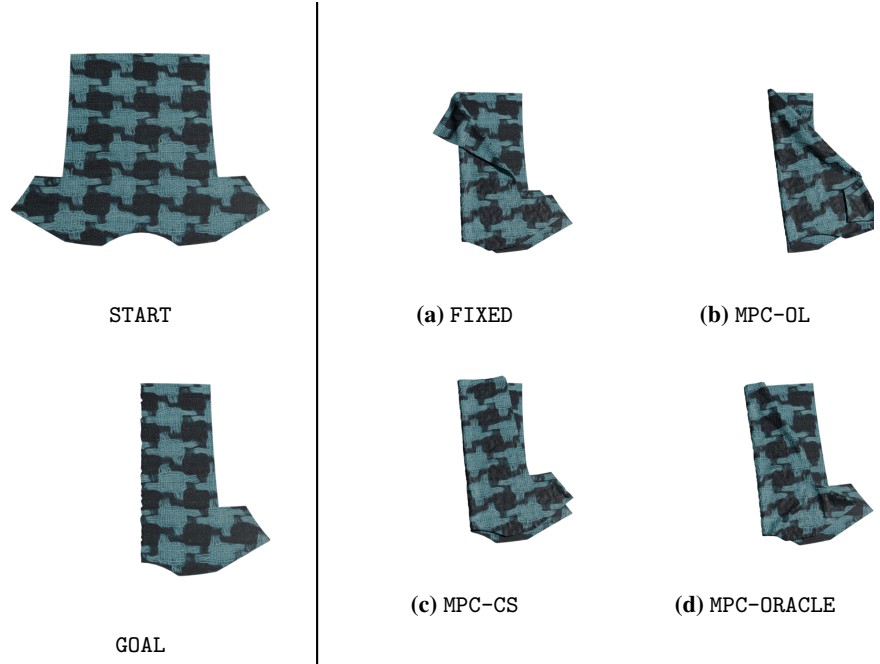

START     **(a)** `FIXED`     **(b)** `MPC-OL`

**(c)** `MPC-CS`     **(d)** `MPC-ORACLE`

GOAL

**Figure 8:** Qualitative results of the half folding for different manipulation strategies.

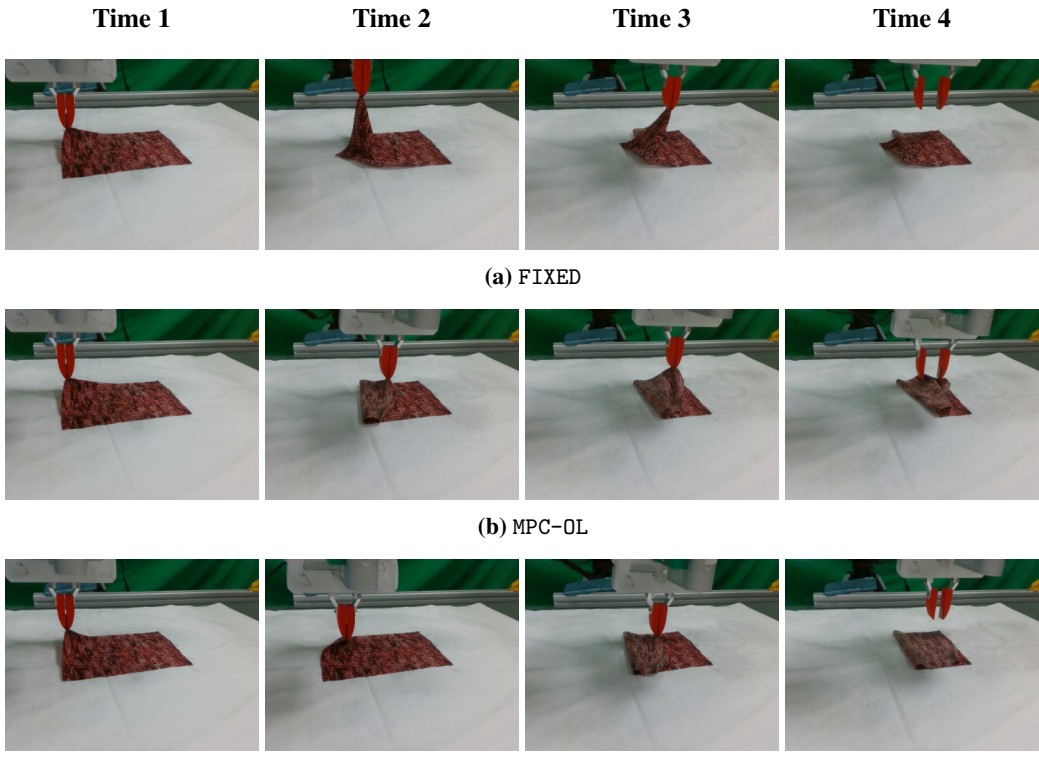

**Figure 9:** Qualitative results of the manipulation outcomes for the methods (a) `FIXED`, (b) `MPC-OL`, and (c) `MPC-CS` are presented. Each method is illustrated at four distinct time points during execution, with the final fold shown at Time 4. Best viewed with zoom.

