# OpenReview forum: "Cloth-Splatting: 3D Cloth State Estimation from RGB Supervision"
_robot-learning.org/CoRL/2024/Conference — CoRL 2024_

### Official Review · Reviewer_wB9e · 2024-07-18
**Initial Review**

**Originality:** 3
**Technical Quality:** 3
**Clarity Of Presentation:** 3
**Potential Impact:** 3
**Recommendation:** 3
**Confidence:** 3

**Review:**

Strength
1. The paper is well written and easy to follow.
2. The constrained GS is an interesting idea. By restricting the GS on the meshes, the proposed method is able to update the states using the differentiable map.
3. The ablation study clearly demonstrates the benefits of each module/loss in the framework.
4. The proposed method outperforms the baselines on the synthetic dataset.

Weakness
1. Since the GS are constrained on the meshes, the performance of the proposed method might be sensitive to the mesh initialization.

2. The explanation on the iterative update is a bit unclear. Is the update parameterized by an MLP with time as the only input (Ln 168?)? Or is it a function of $M$ and $P$? However, in (3) and (4) $\delta M$ is obtained from an optimization? In addition, (3) and (4) should belong to section 4.3, not in the problem statement.

3. Although the proposed method shows great performance using the metrics provided in the paper, the current metrics do not capture the “missing mesh.” For example, in Figure 1 one can clearly see that some part of the cloth is not covered by the mesh. If a method is initialized conservatively, say, only in the middle of the cloth, it can end up having better results because the large deformation is excluded from the evaluation.

**Quality Of The Limitations Section:**

3

**Questions For Rebuttal:**

1. Can the GNN be overfitted to a specific motion? If we commend a motion that GNN has not seen before, will it still have a decent prediction?
2. How sensitive is the proposed method to the initialization of the meshes? Since the GS are constrained on the meshes, I imagine if the meshes are initialized incorrectly, the proposed method might not converge well?
3. How sensitive is the proposed method to the type of motion? Currently in the experiments there is only limited vertical movement of the clothes. Does it still track the clothes well if they are moved vertically?
4. How are $b_1,b_2,b_3$ determined in (8)?
5. What is the predefined distance set for the position accuracy metric?
6. How are the nodes initialized? From the provided video, it looks like there are some empty (transparent) spaces in the cloth. Is it an effect of missing nodes or is it because the GS does not capture the correct color?
7.  In addition, is there a way to prevent incorrect mesh initialization from happening?

**Robotics Focus:**

4

**Summary Of Paper:**

The paper proposes Cloth-Splatting, a deformable object state estimation framework based on a graph neural network (GNN) and a constrained Gaussian splatting (GS). In an iterative process, the state of the object is predicted by the GNN first. Then it’s corrected using the measurement with the constrained GS. The proposed method is evaluated quantitatively on a cloth tracking dataset and qualitatively on a real robot experiment. The cloth tracking experiment shows the proposed method outperforms the baselines.

**Summary Of Recommendation:**

This paper proposes an interesting framework for deformable object state estimation. The constrained GS formulation is novel, and the performance on the synthetic dataset outperforms the baselines. However, the presentation on the state update can be improved, and there remain some questions to be answered. As a result, I’m recommending a weak accept.

---

### Official Review · Reviewer_FCbJ · 2024-07-25
**Good use of GS, needs some refinement**

**Originality:** 4
**Technical Quality:** 4
**Clarity Of Presentation:** 3
**Potential Impact:** 4
**Recommendation:** 4
**Confidence:** 3

**Review:**

The paper presents a well-motivated approach to deformable object state estimation. This work holds significant implications for applications in robotics, where accurate and efficient cloth state estimation is necessary to manipulate them. The experiment on real-world cloth manipulation is particularly encouraging.

The following is some feedback and areas for improvement, including questions to be addressed in the rebuttal.

1. As part of the motivation, you should cite works that currently use visual manipulation of cloth [1][2][3]. Since there isn't much established work on cloth state estimation in the real world.
1. The method requires a complete and occlusion-free prior mesh of the cloth at the initial time step($M0 = (V0, \dot(V0), E0)$), limiting its applicability to scenarios where such a mesh is readily available. You should add this to the limitations.
1. With regards to your stated limitation that regular GS is static. You know you’re using such a GS for trajectories and motions in time. Why not use other GS methods, like those with deformation fields that can account for motion?
1. MD-splatting optim times (line 86-88): What is long per-scene optimization time? What is your per-scene optimization time? Can you include a table with comparisons of these?
1. Line 178: You say that $L_{iso}$ ensures neighboring vertices in cloth maintain a constant distance: Can you explain this? I thought the cloth can stretch or compress, since it is deformable. So triangle edges in the mesh can enlarge or become smaller.
1. Line 220 - 224: When you measure distance between ground truth and predicted cloth meshes, what distance metric do you use?
1. Your choice of a pre-trained dynamics model as the structural graph prior raises questions about its necessity and potential alternatives.
    1. @sim experiments: You have access to a sim, and provide the same inputs to your method as you do to the sim. Using the simulator itself as the structural prior could simplify the approach. So the sim results are good, but not as interesting. The real benefit of your method will show in the real world, when the simulator will fail to appropriately model the cloth deformation (sim2real gap).
    1. @real world: Even for real world expts, you currently use the sim to create dynamics data to train a dynamics model. And during execution, you need a full cloth mesh at time t=0. And robot actions at every timestep. You could just give this data to the simulator and use that as a structured prior. That way you save time and effort by not needing to train the dynamics model.
1. Additionally, since the real world is where the method can truly show its benefits, it would be great to have more expts in the real world with cloth manipulation. In the review, I have marked 'Robotics Focus' as 'Sufficient demonstration on hardware', but at least one more deformable expt would be helpful.

*Minor Points*

1. Typo: $\thickapprox 85$% should be $\sim 85$%, if you mean to say “approx 85%”
1. Typo line 52: What do you mean by 57% times? And ~85% times?
1. Typo line 66: Similarly to our work
1. Typo line 78: NeRF, not NerF
1. Section 3: It’s usually common to use symbol S for state (not M), and symbol O for observation (not Y).
1. Eq 4: Define $\psi$ earlier – it’s used in Eq (4) but defined in line 168
1. Typo line 145: ‘evaluating a the projected 2D Gaussian’
1. Line 238-250: When you say (1), (2) etc – are you referring to Eq. (1), Eq. (2)? At first I thought it was a numbered list. Please clarify.
1. The term "action-conditioned dynamics model" might be unnecessarily verbose, as most dynamics models in robotics and related fields inherently consider states and actions as inputs.

Update: Additional clarifications and real world experiments during the rebuttal are good. Thank you for your contribution.

*References:*

[1] Avigal, Y., Berscheid, L., Asfour, T., Kröger, T., & Goldberg, K. (2022, October). Speedfolding: Learning efficient bimanual folding of garments. In 2022 IEEE/RSJ International Conference on Intelligent Robots and Systems (IROS) (pp. 1-8). IEEE.

[2] Salhotra, G., Liu, I. C. A., Dominguez-Kuhne, M., & Sukhatme, G. S. (2022). Learning deformable object manipulation from expert demonstrations. IEEE Robotics and Automation Letters, 7(4), 8775-8782.

[3] Xu, Z., Chi, C., Burchfiel, B., Cousineau, E., Feng, S., & Song, S. (2022). Dextairity: Deformable manipulation can be a breeze. arXiv preprint arXiv:2203.01197.

**Quality Of The Limitations Section:**

2

**Questions For Rebuttal:**

Please see main review for the feedback and questions that need to be addressed.

**Robotics Focus:**

4

**Summary Of Paper:**

Cloth-splatting addresses the crucial challenge of state estimation in deformable object manipulation, a problem that has traditionally been difficult to solve due to the complex and dynamic nature of deformable materials. Key contributions include mesh-constrained Gaussian splatting for deformables, and an image-based 3D state estimation framework to accurately track deformables like cloth.

**Summary Of Recommendation:**

Cloth-splatting presents a promising solution to the challenging problem of state estimation for deformable objects. The use of gaussian splatting is a great potential benefit to updating cloth meshes for real world experiments. However, addressing the feedback above and providing clarification on certain technical aspects would strengthen the paper's impact and adoption.

---

### Official Review · Reviewer_eUkr · 2024-07-29
**Nice idea well executed, but could use more experiments**

**Originality:** 3
**Technical Quality:** 3
**Clarity Of Presentation:** 3
**Potential Impact:** 3
**Recommendation:** 3
**Confidence:** 3

**Review:**

Pros:
+ I like the idea of using gaussian splatting as a way of correcting for errors based on visual observation. It was novel to me and seems like a good fit for the problem.
+ Incorporating closed loop solutions to this problem seems like a clear benefit over previous approaches
+ Results outperform existing methods on the experiments
+ Ablation analysis shows the benefit of several proposed methods
+ Results generalize to real world examples (in a small sample size)

Cons:
- The experiments were limited to 5 cloth examples with 2 sequences each. This is a pretty small sample size for simulation which could provide infinite samples.
- The methods section was a bit hard to follow at times, but many of these issues could be corrected with some rewrites.
- The problem scope itself is fairly limited to the niche problem of estimating (but not manipulating) cloth. I would have liked to see a method use this estimation to advance some end task performance.

Quality: The research seems well thought-out and backed by mathematical foundations and experiments in sim and real world.
Clarity: I think the writing quality was quite high in general, but certain parts of the method were unclear to me or arranged oddly (see below).
Originality: This method builds on prior work but has interesting novel improvements on those methods.
Significance: I think this work lacks broad significance due to its nature of being applied to the niche problem of cloth pose estimation. Additionally, as the authors mention, the runtime being slower than real-time means this method is probably not useful in actual robotic manipulation tasks involving cloths. However I do think the advancement over previous methods is significant enough to merit interest.

Clarity:
I feel certain rearranging of information would have made this paper easier to understand on a first read-through. Specifically, the placement of equations 1 and 2 before their "english language" explanations made me as a reader do more parsing and rereading than necessary. I would put the explanation first and then end with the equations. Similarly equation 4 takes argmin of psi, but never defines psi until 168 which was confusing to read. Another example is line 121 referencing figure 2 which appears an entire page earlier. As a rule, I would avoid putting a figure well before it is referenced or it will be either read out of context or ignored.

Small typo: Line 52 says 57% times and 85% times. Is it 57% (as I suspect) or 57 times (which would be much much larger).

**Quality Of The Limitations Section:**

3

**Questions For Rebuttal:**

Certain things in this paper were mentioned but never defined in a satisfactory way. I would like the authors to answer some questions about the definitions in their methods section.
- The action space is mentioned several times but never defined. How is it parameterized? Motor torques? What is the dimensionality?
- How is the process initialized? Does it always assume a uniform flat cloth to start? It seems like this method updates existing predictions but wouldn't be able to give an initial prediction. Having just one initialization would severely limit the usefulness of the method.
- Over time, does the GNN incorporate the output of the GN or does it just adjust on a per-timestep basis? The video provided makes it look like the GNN is not changed so over time it accumulates more and more error.
- The normalization constant was given a very long formula but never referenced. Does it matter or is it factored out by way of the minimization performed?
- What range is the integral over in the normalization constant and equation 2? dM_t doesn't mean much to me (or could mean several things).

Additionally I have some questions about the experiments:
- Can the authors explain why RAFT-Oracle does so poorly when it has access to extra information? I would have expected it to outperform other methods, though I'm not entirely sure what RAFT-Oracle actually is. However it seems to severely underperform.
- Why were there so few cloth examples and so few experimental trials? Given the experiments are in simulation, there should be an abundance of examples the authors could use. This in my opinion is the biggest flaw in the paper. It's hard to draw strong conclusions from 10 trials.

**Robotics Focus:**

4

**Summary Of Paper:**

This paper presents a way of using gaussian splatting to correct for errors of a graph neural network representing a cloth simulation

**Summary Of Recommendation:**

The paper has strong foundations, but the lack of sufficient experiments means it could be of limited effectiveness outside the examples tried.

---

### Author Rebuttal · Authors · 2024-08-10

We thank the reviewers (eUkr, FCbJ, wB9e) and the meta reviewer for their constructive feedback and the suggestions for improving the paper. We are glad to hear the reviewers found our method novel and saw the benefit of incorporating a GNN based prediction step with a Gaussian Splatting (GS) based state update.

To answer the questions of the reviewers and to improve the technical quality of our work, we performed additional experiments and included their discussion in rebuttal_experiments.pdf in the attached zip. Specifically, we:
- Extended the quantitative tracking results in simulation by an order of magnitude;
- Evaluated the robustness of our method against incorrect mesh initializations;
- Compared 1) unrolling the GNN for the full episode length and then refining all the predicted state with GS once, against 2) alternating between GNN prediction and GS updates;
- Added manipulation experiments in both simulation and real world to demonstrate the applicability of our method to tasks beyond state estimation.

Additional qualitative tracking results on 5 simulated and 7 real scenes can be found in the folder additional_qualitative_results. Qualitative results for the real world manipulation experiment can be found in the folding_real_world folder.

We further uploaded an updated version of the paper (paper_rebuttal.pdf) which includes a related work section on visual manipulation of cloth, improved problem formulation and method section, and minor changes based on suggestions from all reviewers. New parts are marked in green. Additional experiments from the rebuttal will be included in the camera ready version in case of acceptance.

---

### Decision · Program_Chairs · 2024-09-04

**Decision:**

Accept

**Comment:**

Summarizing the strengths and weaknesses pointed out by the reviewers:

Strengths:
- Using GS for closed-loop error correction of 3D state is novel
- Results outperform existing methods
- Good ablation analysis
- Generalization to real world (at least for a small sample size)

Weaknesses:
- Experiment sample size is fairly small, especially given it's in sim
- Limited to only state estimation, but no manipulation
- Missing some citations to other papers relating to cloth
- Method requires occlusion free initial state of cloth

The reviewers did have differing opinions on a couple points. 1 of the reviewers pointed out some clarity issues in the writing, while another thought it was well-written and clear. This suggests that at least some improvement in the clarity can be made. Another point is at least one reviewer pointed out the nicheness of the problem, while another said it is highly applicable to robotics. The authors might consider better justifying why robotics should care about 3D cloth state estimation.

Overall, all 3 reviewers recommend acceptance, although weakly. The authors should pay special attention to the specific concerns brought up by each reviewer to increase the likelihood the paper is accepted.

**Update after rebuttal**: The authors' rebuttal was quite extensive. They directly addressed many of the reviewers' concerns, and even ran several more experiments. All 3 reviewers responded to the rebuttal, 2 deciding to keep their weak accept recommendation and 1 reviewer upgrading to strong accept. I think this is a clear indication that this paper is a significant contribution to the field, and should be accepted.